Methods

# Reconstructing B-cell receptor sequences from short-read single-cell RNA sequencing with BRAPeS

Shaked Afik[1,*], Gabriel Raulet[2,*], Nir Yosef[1,3,4,5]

RNA sequencing of single B cells provides simultaneous measurements of the cell state and its antigen specificity as determined by the B-cell receptor (BCR). However, to uncover the latter, further reconstruction of the BCR sequence is needed. We present BRAPeS ("BCR Reconstruction Algorithm for Paired-end Single cells"), an algorithm for reconstructing BCRs from short-read paired-end single-cell RNA sequencing. BRAPeS is accurate and achieves a high success rate even at very short (25 bp) read length, which can decrease the cost and increase the number of cells that can be analyzed compared with long reads. BRAPeS is publicly available at the following link: https://github.com/YosefLab/BRAPeS.

## Introduction

B cells play a significant role in the adaptive immune system, providing protection against a wide range of pathogens. This diversity is due to the B-cell receptor (BCR), which enables different cells to bind different pathogens (1). Single-cell RNA sequencing (scRNA-seq) has emerged as one of the leading technologies to characterize and study heterogeneity in the immune system across cell types, development, and dynamic processes (2, 3). Combining transcriptome analysis with BCR reconstruction in single cells can provide valuable insights to the relation between BCR and cell state, as was demonstrated by similar studies in T cells (4–6).

The BCR comprises two chains, a heavy chain and a light chain (either a κ or λ chain). Each chain is encoded in the germline by multiple segments of three types—variable (V), joining (J), and constant (C) segments (the heavy chain also includes a diversity [D] segment, see the Materials and Methods section). The specificity of the BCRs comes from the V(D)J recombination process, in which for each chain, one variable (V) and one joining (J) segment are recombined in a process that introduces insertions and deletions into the junction region between the segments, called the complementarity-determining region 3 (CDR3) (7). The resulting sequence is the main determinant of the cell's ability to recognize a specific antigen. After B-cell activation, somatic hypermutations are introduced to the BCR, and the constant region may be replaced in a process termed isotype switching (8). The random mutations make BCR reconstruction a challenging task. Although methods to reconstruct BCR sequences from full-length scRNA-seq are available (9–11) (as well as single-cell V(D)J-enriched libraries from 10x Genomics: https://www.10xgenomics.com/solutions/vdj/), they were only tested on long reads (150 and 50 bp). The ability to reconstruct BCR sequences from short (25–30 bp) reads is important, as it can decrease cost which can, in turn, increase the number of cells that could be feasibly analyzed.

We introduce BRAPeS ("BCR Reconstruction Algorithm for Paired-end Single cells"), an algorithm and software for BCR reconstruction. Conversely to other methods, BRAPeS was designed to work with short (25–30 bp) reads, and indeed we demonstrate that under these settings, it performs better than other methods. Furthermore, we show that the performance of BRAPeS when provided with short reads is similar to what can be achieved with much longer (50–150 bp) reads from the same cells, suggesting that BCR reconstruction does not necessitate costly sequencing with many cycles.

## Results

BRAPes is an extension of the TCR reconstruction software TRAPeS (4), with significant modifications added to address the processes of isotype switching and somatic hypermutations, which are specific to B cells (Fig 1, see the Materials and Methods section for full description of the algorithm). Briefly, BRAPeS takes as input the alignment of the reads to the reference genome. BRAPeS first recognizes the possible V and J segments by finding reads with one mate mapping to a V segment and the other mate mapping to a J segment. All unmapped reads whose mates were mapped to the V/J/C segments are then collected, assuming that

[1]Center for Computational Biology, University of California, Berkeley, Berkeley, CA, USA [2]Department of Computer Science, University of California, Davis, Davis, CA, USA [3]Department of Electrical Engineering and Computer Science, University of California, Berkeley, Berkeley, CA, USA [4]Ragon Institute of Massachusetts General Hospital, Massachusetts Institute of Technology and Harvard, Cambridge, MA, USA [5]Chan Zuckerberg Biohub, San Francisco, CA, USA

Correspondence: niryosef@berkeley.edu
*Shaked Afik and Gabriel Raulet contributed equally to this work

Identify V and J segments based on alignment to the reference genome

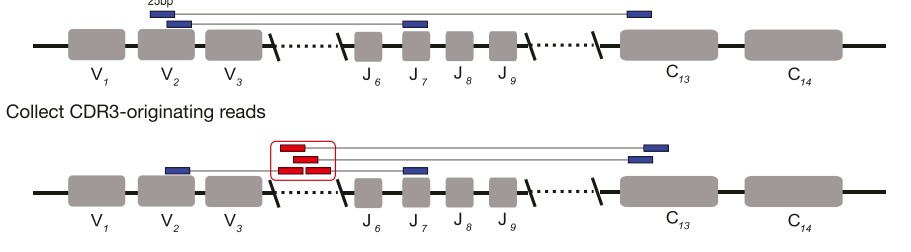

Collect CDR3-originating reads

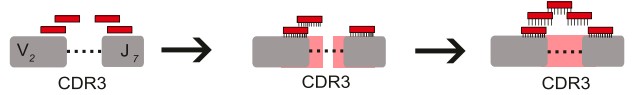

CDR3 reconstruction

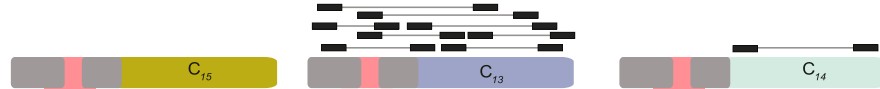

Isotype determination

Somatic hypermutation correction

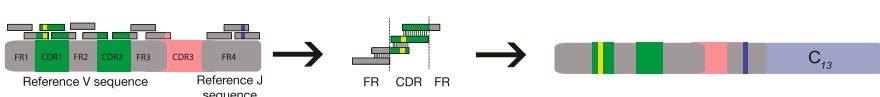

**Figure 1.  The BRAPeS algorithm.**
First, the V and J segments are selected based on the initial alignment to the reference genome by searching for paired reads with one read mapping to a V segment and its mate mapping to a J segment. Next, putative CDR3-originating reads are identified as the unmapped reads whose mates map to the V/J/C segments. BRAPeS runs an iterative dynamic programming algorithm to align the CDR3-originating reads to the V and J segments and extend them until they overlap. BCR isotype is then determined by running RSEM on all possible full BCR transcripts (the reconstructed V–J segments combined with all possible constant segments). Finally, BRAPeS corrects for somatic hypermutations by building a consensus sequence of the reads aligning to the CDR1, CDR2, and FRs.

most CDR3-originating reads will be unmapped when aligning to the reference genome. Then, the CDR3 region is reconstructed with an iterative dynamic programming algorithm. At each step, BRAPeS aligns the unmapped reads to the edges of the V and J segments, using the sequence of the aligned reads to extend the V and J sequences until convergence. Next, the BCR isotype is determined by appending all possible constant segments to the reconstructed sequence and taking the most likely complete transcript based on transcriptomic alignment with RSEM (12). Finally, BRAPeS corrects for somatic hypermutations by collecting all reads aligning to the genomic regions of the CDR1, CDR2, and the framework regions (FRs) and aligning these reads against each other to obtain a reconstruction of the consensus sequence. The CDR3 sequences and their productivity are determined based on the criteria established by the international ImMunoGeneTic (IMGT) information system (13,14) (see the Materials and Methods section).

We evaluated BRAPeS' performance on 374 cells from two previously published datasets—174 human B cells and 200 mouse B cells (see the Materials and Methods section and Table S1) (9,15). To evaluate BRAPeS, we first trimmed the original reads (50 bp for the human data and 150 bp for the mouse data) and kept only the outer 25 or 30 bases. We compared BRAPeS' performance on the trimmed data to two other previously published methods—BASIC (9) and VDJPuzzle (10) applied either on the trimmed data or the original long reads.

When applied to 30-bp reads, BRAPeS' success rates are similar to other methods for the light chain but are higher for heavy chain reconstruction (Fig 2A and Table S2). BRAPeS reconstructs productive heavy chains in a total of 348 cells, 93% of the cells across both datasets and reconstructs productive light chains in 370 cells (98.9% of the cells). These results are in line with the success rates of BASIC and VDJPuzzle on the original long reads: BASIC reconstructs productive heavy and light chains in 353 (94.4%) and 364 (97.3%) cells, respectively, and VDJPuzzle reconstructs heavy chains in 346 (92.5%) cells and light chains in 368 (98.4%) cells. On 30-bp reads, BASIC and VDJPuzzle achieve similar reconstruction rates for the light chain (362 [96.8%] cells and 370 [98.9%] cells with a productive light chain in BASIC and VDJPuzzle, respectively). However, BASIC and VDJPuzzle see a decline in success rates for the heavy chain, reconstructing a productive heavy chain in only 273 (73%) cells for BASIC and 242 (64.7%) cells for VDJPuzzle (Fig 2A and Table S2).

BRAPeS is also able to maintain a high success rate on 25-bp reads, reconstructing heavy chains in 328 (87.7%) cells and light chains in 370 (98.9%) cells (Fig 2B and Table S3). Yet, we observe a substantial decrease in the results of other methods. VDJPuzzle is unable to reconstruct any chains with 25-bp reads. This is likely due to its use of the de-novo assembler Trinity (16), which requires a seed k-mer length of 25 bp that is unsuitable for very short reads. Similarly to 30 bp, BASIC is able to maintain a high reconstruction rate for light chains, with productive reconstructions in 363 (97.1%)

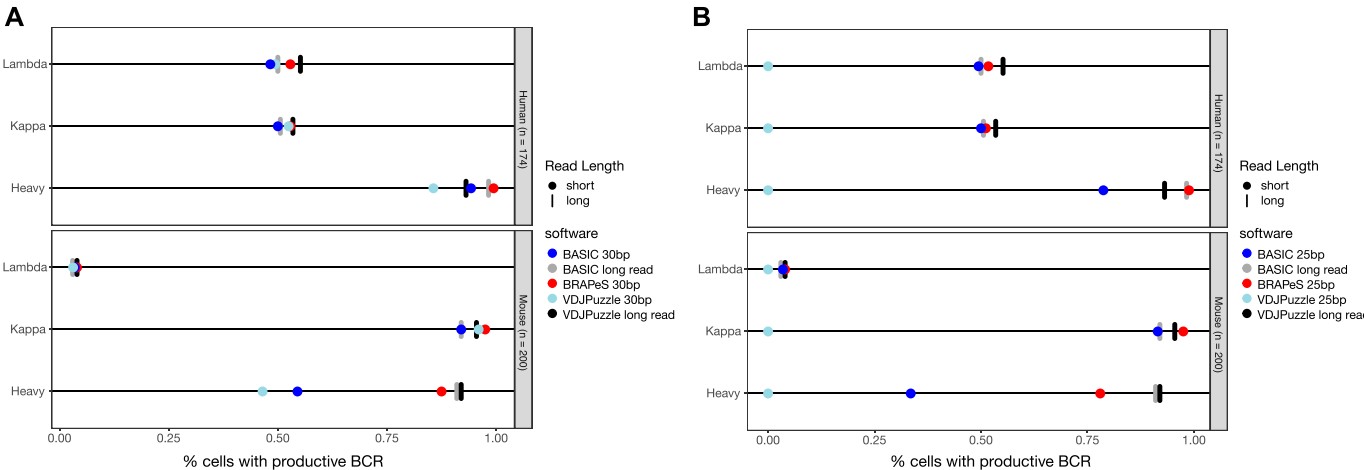

**Figure 2. BRAPeS success rates.**
**(A)** Fraction of cells with a successful reconstruction of a productive CDR3 in human and mouse B cells using the following methods: VDJPuzzle applied to the original, long-read data (black line), and the trimmed version of the data, trimmed to 30 bp (light blue circle). BASIC applied to the long-read (grey line) and the trimmed data (dark blue circle), and BRAPeS applied to the trimmed data (red circle). **(B)** Same as (A), but the trimmed version of the data was trimmed down to include only the outer 25 bp, instead of 30 bp.

cells, but is only able to reconstruct productive heavy chains in 204 (54.5%) cells (Fig 2B and Table S3). Moreover, BASIC only outputs fasta sequences, thus requiring further processing to annotate the BCR.

We next turn to evaluate the accuracy of the short-read–based CDR3 reconstructions, by comparing the resulting sequences to those obtained with long reads (Fig 3 and see the Materials and Methods section). We use the long-read–based reconstruction of BASIC as a reference (we achieve similar results with VDJPuzzle on the long-read data; see Fig S1) and evaluate the accuracy in terms of sensitivity (how many of the CDR3 sequences in the full-length data have an identical reconstruction with the short-reads) and specificity (how many of the CDR3 sequences in the short-read data have an identical long-read reconstruction). In general, all methods show a high level of specificity, having almost all CDR3 sequences identical to the sequences reconstructed on long reads, whenever both read lengths produce a productive reconstruction (Fig 3A and B). In accordance with the higher success rate, BRAPeS shows a high sensitivity, with a rate of 0.96 for 30-bp data and 0.92 for 25-bp data (Fig 3C and D). This is in line with the agreement of different methods on the original data, as VDJPuzzle on long reads has a sensitivity rate of 0.96. On the trimmed data, BASIC and VDJPuzzle show a lower sensitivity rate—BASIC achieves sensitivity rates of 0.87 and 0.78 for 30- and 25-bp, respectively, and VDJPuzzle has a sensitivity rate of 0.83 for 30 bp. These results also hold if we only take the top-ranking reconstruction of BRAPeS, as more than 97.5% of the identical CDR3 sequences between BRPAeS and BASIC are the highest ranked sequences for both 25 and 30 bp (Figs S2 and S3).

BRAPeS' correction of somatic hypermutations is also accurate across the various regions of the transcript (Fig 3). Besides a slight decrease in specificity for CDR2 and FR1 reconstruction, BRAPeS maintains a very high level of specificity across all regions in line with the other methods. We note that BASIC achieves lower specificity rates for FR1 reconstructions for short reads mostly because of partial reconstructions. Overall, BRAPeS has a high

sensitivity rate across all regions (0.92–0.97 for 30 bp and 0.88–0.94 for 25 bp), comparable with the sensitivity of VDJPuzzle on long reads (0.87–0.95). Similar to the CDR3 results, the high sensitivity and specificity hold when comparing only the top-ranking reconstruction, as identical regions are 96.8–99.9% of the top-ranking regions for 30 bp and 95.4–100% of the top-ranking regions for 25 bp (Figs S2 and S3).

## Discussion

Coupling BCR reconstruction with transcriptome analysis in single cells can provide valuable information about the effect of antigen specificity and isotype to cellular heterogeneity. Despite an increase in technical noise in transcriptome analysis compared with longer reads (17,18), short-read sequencing is still widely used as it can reduce sequencing costs by hundreds to thousands of dollars per run, depending on the sequencing platform and desired total number of reads. However, current methods do not provide a sufficient solution for reconstructing immune cell receptors from short reads (18). To this end, we provide BRAPeS, a software for BCR reconstruction tailored to work on short-read scRNA-seq. BRAPeS is accurate and has a success rate on short reads similar to other methods applied to long reads, demonstrating that BCR reconstruction can be achieved at a much lower cost. BRAPeS is publicly available at https://github.com/YosefLab/BRAPeS.

## Materials and Methods

### The BRAPeS algorithm

The input given to BRAPeS is a directory where each subdirectory includes genomic alignments of a single cell.

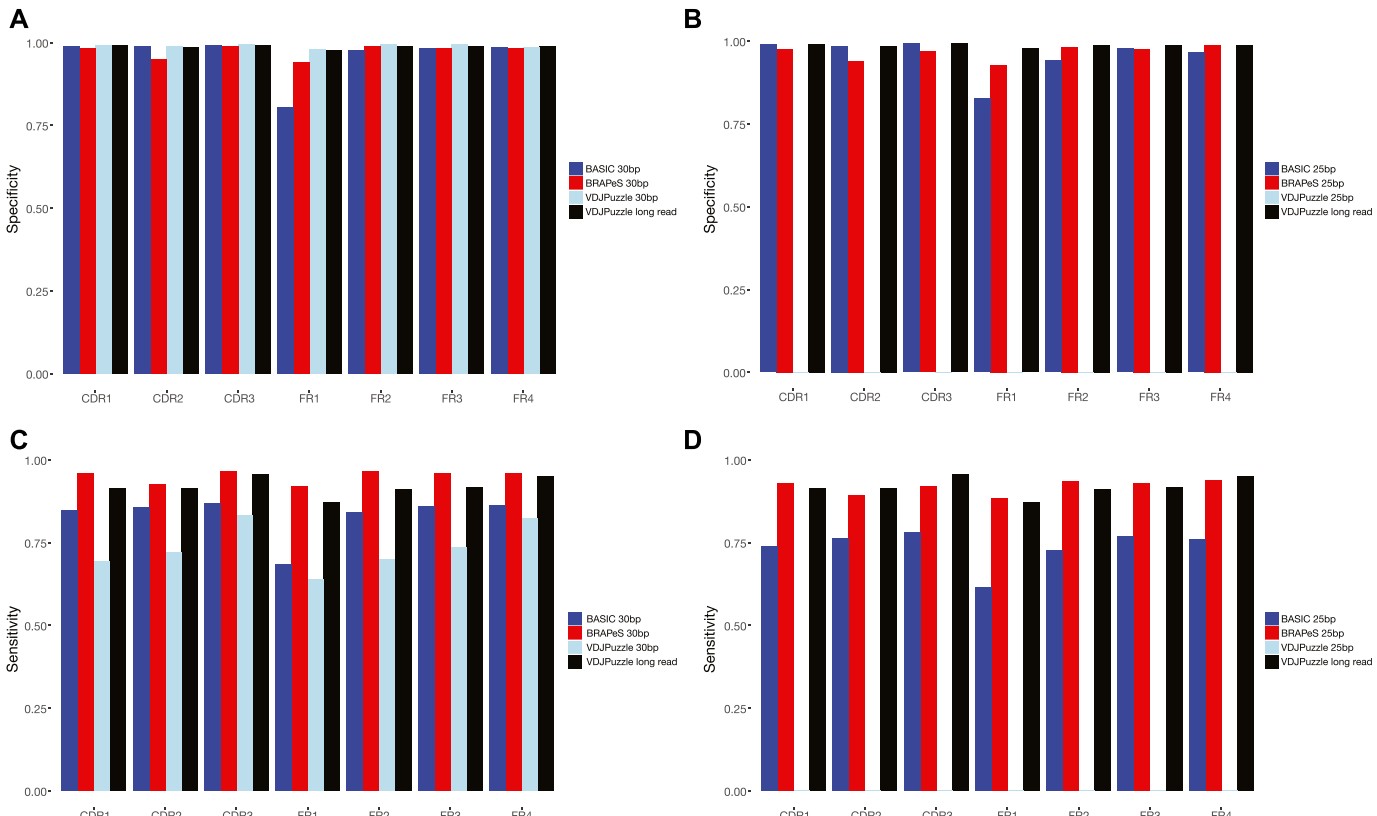

**Figure 3. Sensitivity and specificity of BRAPeS.**
**(A)** Specificity of BRAPeS for 30 bp for each CDR and FR. The fraction of chains with a sequence identical to the sequence reconstructed by BASIC on the long-read data for each region, using the following methods: VDJPuzzle when applied to the long-read data (black), BRAPeS (red), BASIC (dark blue), and VDJPuzzle (light blue) applied to a version of the data trimmed to 30 bp. The fraction is calculated only for chains that had a productive reconstruction in both the long-read BASIC results and the other method. **(B)** Specificity of BRAPeS for 25 bp. Same as (A), except the short-read version of the data was trimmed to include only the outer 25 bp, instead of 30 bp. **(C)** Sensitivity of BRAPeS for 30 bp for each CDR and FR. Same as (A), except the fraction is calculated out of all the chains that had a productive reconstruction when running BASIC on the long-read data. **(D)** Sensitivity of BRAPeS for 25 bp. Same as (B), except the fraction is calculated out of all the chains that had a productive reconstruction when running BASIC on the long-read data.

The BRAPeS algorithm has several steps, performed separately for each chain in each cell:

1. **Identifying possible pairs of V and J segments**: BRAPeS searches for reads where one mate of the pair is mapped to a V segment and the other mate is mapped to a J segment. BRAPeS collects all possible V–J pairs and attempts to reconstruct complete BCRs from all possible pairs. Because the D segment is very short, reads do not align to it; thus as part of the reconstruction step (step 3), the sequence of the D segment is also reconstructed. If no V–J pairs are found, BRAPeS will look for V–C and J–C pairs and will take all possible V/J pairing of the found V and J segments.

   In case of many possible V–J pairs (which can occur because of the similarity among the segments), the user can limit the number of V–J pairs to attempt reconstruction on. BRAPeS will rank the V–J pairs based on the number of reads mapped to them and take only the top few pairs (the exact number is a parameter controlled by the user).

2. **Collecting the set of putative CDR3-originating reads**: BRAPeS collects the set of reads that are likely to originate from the CDR3 region. Those are the reads that are unmapped to the reference

genome, but their mates are mapped to the V/J/C segments. In addition, because the first step of CDR3 reconstruction includes alignment to the ends of the genomic V and J sequences, reads mapping to the V and J segments are also collected.

3. **Reconstructing the CDR3 region**: For each V–J pair, the edges of the V and J segments are extended with an iterative dynamic programming algorithm. In each iteration, BRAPeS tries to align all the unmapped reads to the V and J sequences separately with the Needleman–Wunsch algorithm with the following scoring scheme: +1 for match, −1 for mismatch, −20 for gap opening, and −4 for gap extension. In addition, BRAPeS does not penalize having a read "flank" the genomic segment. All reads that passed a user-defined threshold are considered successful alignments. BRAPeS then builds the extended V and J segments by taking for each position the base which appears in most reads. This process repeats for a given number of iterations or until the V and J segments overlap. Because the purpose of this step is to reconstruct only the CDR3 region, to reduce running time the alignment is performed only on a predetermined number of bases leading to the ends of the V and J segment (3′ end of the V segment and 5′ end of the J segment). The number of bases taken from the end of each segment is a parameter controlled by the user, set by

default to the length of the J segment. BRAPeS can also run a "one-sided" mode, where if an overlap was not found (e.g., because of assigning the wrong V segment), BRAPeS will attempt to determine the productivity of only the extended V and only of the extended J segment.

4. **Isotype determination**: To find the BCR isotype, for each V–J pair with a reconstructed CDR3, BRAPeS concatenates the full sequences of all possible constant segments. Then, BRAPeS runs RSEM (12) on all sequences using all paired-end reads with at least one mate mapped to the genomic V/J/C segments as input. For each V–J pair, the constant region with the highest expected count is taken as the chosen constant segment.

5. **Somatic hypermutation correction**: All the reads from step 4 are aligned against the genomic CDR1, CDR2, and framework sequences obtained from IMGT using the SeqAn package (19). Reads are chosen as candidates for reconstruction if the percentage of mutations in the aligned sequence is below a given input threshold, set by default to 0.35 for CDRs and 0.2 for FRs. Separate thresholds are used for framework and CDRs to account for higher rates of somatic hypermutations in the CDRs. When reads align across adjacent CDR-FRs, the rate of mutation is calculated separately for the aligned framework segment and the aligned CDR segment. If both score below their given thresholds, the read is saved for reconstruction. Once all putative reads have been collected, they are first aligned based on the coordinates obtained from the genomic alignments. Then, to correct for possible misalignments, the consensus alignment algorithm in the SeqAn package is run using these approximate positions as guides. Finally, the reconstructed sequence is obtained by aligning the genomic sequences against the consensus sequence to find their start and end coordinates.

6. **Separating similar BCRs and determining chain productivity**: After selecting the top isotype for each V–J pair and correcting for somatic hypermutations, BRAPeS determines whether the reconstructed sequence is productive (i.e., the V and J are in the same reading frame with no stop codon in the CDR3) and annotates the CDR3 junction. If more than one V–J pair produces a CDR3 sequence (either because of having more than one recombined chain in the cell or because of similar V–J segments resulting in the same CDR3 sequence reconstruction), the various productive reconstructions are ranked based on their expression values as determined by RSEM.

The output for BRAPeS is the full ranked list of reconstructed chains, including the CDR3 sequences, V/J/C annotations and the number of reads mapped to each segment, as well as a summary file of the success rates across all cells. In addition, for each cell, the output is the full sequence of each reconstructed BCR, as well as a file detailing the sequences of the CDR1, CDR2, and FRs, a file with the read count for each isotype and a file with the read count for each productive BCR.

BRAPeS is implemented in python. To increase the performance, the dynamic programming algorithm and the somatic hypermutation correction algorithm is implemented in C++ using the SeqAn package (19). Moreover, to decrease the running time for deeply sequenced cells, BRAPeS has the option to randomly downsample the number of reads for CDR3 reconstruction to 10,000 and the number of reads for somatic hypermutation correction to

40,000. BRAPeS is publicly available and can be downloaded at the following link: https://github.com/YosefLab/BRAPeS.

## Data availability and preprocessing

Raw fastq files of mouse B cells were downloaded from Wu et al (ArrayExpress E-MTAB-4825) (15). All analyses were performed on the 200 cells that were available through ArrayExpress. Raw fastq files for the human data from Canzar et al (9) were provided by the author. We excluded single-end cells and cells filtered out in the original study, leaving a total of 174 cells. Next, the reads were trimmed to be 25- or 30-bp paired-end with trimmomatic (20), keeping only the outer bases.

For BRAPeS, low-quality reads were trimmed using trimmomatic with the following parameters: LEADING:15, TRAILING:15, SLIDINGWINDOW:4:15, and MINLEN:16. The remaining reads were aligned to the genome (hg38 or mm10) using Tophat2 (21). Running VDJPuzzle and BASIC on the reads after quality trimming resulted in no reconstructions for VDJPuzzle and a slight decrease in reconstruction rates for BASIC; thus, the results presented in the article for VDJPuzzle and BASIC are for the raw reads.

## Running BRAPeS

For this study, BRAPeS was run using the following parameters for the human data: "-score 15 -top 6 -byExp -iterations 6 -downsample -oneSide." The "score" is the minimal alignment score for the CDR3 reconstruction step and "iterations" limits the number of times BRAPeS attempts to extend the V and J segments. The parameters "top" and "byExp" determine the maximal number of V–J pairs per chain on which reconstruction is attempted, by ranking the pairs based on their number of aligned reads and sampling from the pairs with the highest read count. The "downsample" parameter reduces the number of reads used for CDR3 reconstruction and somatic hypermutation correction.

For the mouse data, BRAPeS was run with the following parameters: "-score 15 -oneSide -byExp -top 10." In addition, as some cells required a higher alignment score threshold, we ran BRAPeS with a scoring threshold of 21 for chains without a productive reconstruction (Table S1).

## Running VDJPuzzle and BASIC

We ran VDJPuzzle using default parameters, providing VDJPuzzle with the hg38 genome and GRCh38.p2 annotation for human, and mm10 genome with the GRCm38.p4 annotation for mouse. We then considered only reconstructions with a complete CDR3 (no missing bases) that appeared in the "summary_corrected" folder as valid productive reconstructions.

BASIC was ran with default parameters. After running BASIC, we collected all the output fasta files and ran them through IMGT/HighV-Quest (22,23). Only sequences that resulted in productive CDR3 according to IMGT were considered successful reconstructions.

## Comparison of sensitivity and specificity

To determine the accuracy of the methods, we compared the reconstructed CDR3 nucleotide sequences with the reconstruction produced by running BASIC or VDJPuzzle on long reads. Only CDR3s with sequences

identical to the sequences reconstructed on the long-read data were considered accurate. In case of more than one reconstructed CDR3 sequence, if both methods had at least one identical CDR3 sequence it was considered an accurate reconstruction, except for Figs S2 and S3, for which we only compared the highest ranking reconstruction. We used the same criteria of a perfect match to estimate the reconstruction accuracy of CDR1, CDR2, FR1, FR2, and FR3 regions. The annotated FR4 VDJPuzzle output was much longer than that of BASIC; thus, when comparing with BASIC, we considered the FR4 sequence accurate if the FR4 prefix was identical to the full BASIC FR4 reconstruction.

## Supplementary Information

### Author Contributions

S Afik: conceptualization, software, formal analysis, methodology, and writing—original draft, review, and editing.
G Raulet: software, formal analysis, and methodology.
N Yosef: conceptualization, supervision, funding acquisition, methodology, and writing—original draft, review, and editing.

### Conflict of Interest Statement

The authors declare that they have no conflict of interest.

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
