## [Reviewer comments · Life Science Alliance]

Life Science Alliance

Reconstructing B cell receptor sequences from short-read single cell RNA-sequencing with BRAPeS

Shaked Afik, Gabriel Raulet, and Nir Yosef
DOI: <https://doi.org/10.26508/lsa.201900371>

Corresponding author(s): Nir Yosef, UC Berkeley

Review Timeline:	Submission Date:	2019-03-08
	Editorial Decision:	2019-04-29
	Revision Received:	2019-08-04
	Editorial Decision:	2019-08-11
	Revision Received:	2019-08-13
	Accepted:	2019-08-14

Scientific Editor: Andrea Leibfried

Transaction Report:

April 29, 2019

Re: Life Science Alliance manuscript #LSA-2019-00371-T

Prof Nir Yosef
UC Berkeley

Dear Dr. Yosef,

Thank you for submitting your manuscript entitled "Reconstructing B cell receptor sequences from short-read single cell RNA-sequencing with BRAPeS" to Life Science Alliance. The manuscript was assessed by expert reviewers, whose comments are appended to this letter. Please excuse the delay in getting back to you - a third report on your work is pending, but since the reviewer did not submit the report despite our chasers, we now decided to move forward without the report.

As you will see, the two reviewers appreciate your work, and they provide constructive input on how to further strengthen the manuscript to increase the value of the tool for the community. We would thus like to invite you to submit a revised version, addressing the points raised by the reviewers. The requests made seem straightforward to address, but please get in touch in case you would like to discuss individual revision points further.

You will be guided to complete the submission of your revised manuscript and to fill in all necessary information. Please contact us in case you do not know or remember your login name.

Thank you for this interesting contribution to Life Science Alliance. We are looking forward to receiving your revised manuscript.

Sincerely,

Andrea Leibfried, PhD
Executive Editor
Life Science Alliance

Meyerhofstr. 1
69117 Heidelberg, Germany
t +49 6221 8891 502
e a.leibfried@life-science-alliance.org
www.life-science-alliance.org

B. MANUSCRIPT ORGANIZATION AND FORMATTING:

Reviewer #1 (Comments to the Authors (Required)):

The manuscript reports of a novel tool, BRAPeS, which allows BCR reconstruction from short reads, namely less than 30bp. BRAPeS utilise a similar approach to other existing software tools, which have been used to obtain a benchmarking comparison. The novelty of BRAPeS is to avoid de novo reconstruction methods and introduce iterative dynamic programming algorithm, which effectively allows to identify heavy and light chains without using k-mers of a minimum length, hence allowing accurate identification of VDJ recombination and other CDR regions. The manuscript is well written and clear to the point, showing how for short reads BRAPeS performed significantly better than two

other tools, VDJPuzzle and BASIC. The comparison with longer reads also show very similar results to the other tools. Overall I found this work suitable for publication and giving an interesting message to the researchers interested in immune receptor analysis.

My only minor comment is to add some more details in the Discussion. Short reads are likely to remain a cost-effective way to perform NGS analysis on single cell data in the near future. However, technologies are improving in terms of cost and read length, and we know that longer reads improves quality of transcriptomics data in general. This point should at least be mentioned in the Discussion. Also, the authors correctly point out that for BCR and TCR reconstruction read length could be reduced. However, the gene expression profile can be significantly impaired by increasing signal to noise ratio, see for example several other analyses performed on bulk RNAseq Conesa, A. et al. A survey of best practices for RNA-seq data analysis. Genome Biol, Changawala et al. The impact of read length on quantification of differentially expressed genes and splice junction detection. Genome Biol 16, as well as Rizzetto et al Scientific Report 2018 on scRNA-seq directly on T cells.

On another positive note, I am wondering whether this approach could also improve the TCR and BCR detection with the 5' kit from 10X and similar? I have seen several works highlighting that the alignment of the reads from these experiments may be of low quality with standard methods (especially those provided by the companies selling the kits). Perhaps the authors can comment on this as future direction? This tool may be adapted to perform more flexible analyses with a range of technologies?

Reviewer #2 (Comments to the Authors (Required)):

1.

The authors present the computational tool BRAPeS, an extension of TRAPeS, for reconstruction of B cell receptors (BCRs) from very short reads (25-30 bp). BRAPeS identifies V/J pairs and then reconstructs the CDR3 region in an iterative fashion. The paper demonstrates that BRAPeS is able to reconstruct the CDR3 regions of BCRs from very short reads in a highly sensitive manner, thus opening up the possibility of performing single-cell RNA-sequencing experiments of B-lineage cells using shorter reads than what are currently being used (50-150 bp). Decreased sequencing costs or an increase in number of cells that could be included in a sequencing experiment for the same cost would be very useful for the community.

2.

The data presented support the claim that BRAPeS is more sensitive than existing methods when using very short reads, especially for 25 bp reads and for the heavy chain. Did the authors try to run BASIC and VDJPuzzle using quality-trimmed reads, as they did for BRAPeS, and if so, whether this resulted in any changes in the outcome?

The evidence provided by the authors that BCR reconstruction by BRAPeS is accurate is not fully convincing. Firstly, the authors only compared the CDR3 sequences reconstructed by the various methods, and did not show any data on the accuracy of the remaining parts of the BCR sequences and/or allele assignments. Secondly, the authors only compared the amino acid sequences of the CDR3s, while this could have easily been performed on the nucleotide level. The accuracy of BCR reconstruction could be better judged by aligning the entire reconstructed BCR nucleotide sequences to the long-read reconstructed BASIC sequences. Furthermore, in the Materials and methods section it is stated that «In case of more than one reconstructed CDR3 sequence, if both

methods had at least one identical CDR3 sequence it was considered an accurate reconstruction». It would be useful to know how many such cases there were, and whether the identical CDR3 was the most highly ranked by BRAPeS. It should be easy to include these aspects in the comparison of sensitivity and specificity between the methods.

While the authors state that BRAPeS corrects for somatic hypermutations in the CDRs, on the GitHub page it is stated that this is currently only true for heavy chain. When will this be implemented for light chains? Furthermore, a major drawback of the method is that it does not seem to correct for somatic hypermutations in the framework regions. If the FWR sequences are determined by the assigned V allele, this could be incorrect both due to incorrect assignment of V allele and due to presence of mutations or polymorphisms. Thus, downstream analyses of the reconstructed BCRs may be based on incorrect BCR reconstructions and mutational profiling analyses may be inaccurate. It is also unclear if J segments are corrected for somatic hypermutations, or if they are reconstructed together with the CDR3 after identification of the V/J pair.

The authors provide no estimations or examples of cost reduction or increase in cell numbers that could be analyzed when reducing the read length from e.g. 50 bp (for which current BCR reconstruction methods work fine) to 25 bp. Very few scRNA-seq studies currently use 25-30 bp reads, and therefore some evidence on whether shorter reads would influence mapping rate and transcript quantification of the remaining transcriptome need to be provided by the authors.

3. Lastly, indicate any additional issues you feel should be addressed (text changes, data presentation, statistics etc.).

P=paragraph number in section, L=line

- The authors should provide a small test dataset and expected output files in order to test a working installation of BRAPeS
- The authors should provide information regarding versions of software dependencies that have been tested with BRAPeS
- Abstract:
 - o The sentence about measurement of binding specificity needs rewriting since the binding specificity per se is not measured. It should be specified that you mean BCR-mediated specific binding.
 - o BRAPeS is publically available at the following link
- Introduction:
 - o P2 L1: The term IgL is potentially confusing to the reader, as it could be interpreted as the lambda chain
 - o P2 L5: "joining segments" should be "joining segment".
 - o P2 L6: The term "mutations" should be removed as this is covered by insertions and deletions, and may be confused with somatic hypermutations
 - o P2 L10: Somatic hypermutations also occur in the framework regions, not only CDRs. Furthermore, class switching does not necessarily occur after activation, so "may be replaced" is a better term.
 - o The sentence mentioning 10x genomics looks very messy with a lot of parentheses
 - o Last sentence: Remove "length" from "short length reads"
- Results:
 - o P1: "BRAPeS" is mentioned very often. Please consider to change some of the sentences to passive voice, omitting BRAPeS.
 - o P2: Typo: BRAPes
- Materials and methods:
 - o The BRAPeS algorithm

- has several steps
- 3. If the number of bases is the length of the J segment, then won't the iteration in effect start at the 3' end of J and not 5' end? It is unclear to me what is meant by the number of bases in this paragraph.
- 4.
- Change BCR class to isotype
- Is the CH1 part of C gene used or the entire constant region gene?
- mapped to the
- What about naïve B cells expressing both IgD and IgM?
- 5. ... optionally extended: Does this mean that the FWRs are optionally reconstructed?
 - o Running BRAPeS: Please provide a brief explanation of the parameters that were used.
 - o The BRAPeS algorithm
- Figure 1: It is confusing to show alignment to segments in the germline configuration when this is not how the locus looks like in the cells being sequenced. It will be easier to understand if V3 and J6 are not shown, or if it is clearly stated that the segments are in germline configuration. Please change "class determination" to "isotype determination"
- Figure 2: Typo, «TCR» should be «BCR»

Editorial comments:

As you will see, the two reviewers appreciate your work, and they provide constructive input on how to further strengthen the manuscript to increase the value of the tool for the community. We would thus like to invite you to submit a revised version, addressing the points raised by the reviewers. The requests made seem straightforward to address, but please get in touch in case you would like to discuss individual revision points further.

We are encouraged by these reviews, and addressed them, as listed below.

Reviewer 1:

The manuscript reports of a novel tool, BRAPeS, which allows BCR reconstruction from short reads, namely less than 30bp. BRAPeS utilise a similar approach to other existing software tools, which have been used to obtain a benchmarking comparison. The novelty of BRAPeS is to avoid de novo reconstruction methods and introduce iterative dynamic programming algorithm, which effectively allows to identify heavy and light chains without using k-mers of a minimum length, hence allowing accurate identification of VDJ recombination and other CDR regions. The manuscript is well written and clear to the point, showing how for short reads BRAPeS performed significantly better than two other tools, VDJPuzzle and BASIC. The comparison with longer reads also show very similar results to the other tools. Overall I found this work suitable for publication and giving an interesting message to the researchers interested in immune receptor analysis.

We thank the reviewer for these remarks.

My only minor comment is to add some more details in the Discussion. Short reads are likely to remain a cost-effective way to perform NGS analysis on single cell data in the near future. However, technologies are improving in terms of cost and read length, and we know that longer reads improves quality of transcriptomics data in general. This point should at least be mentioned in the Discussion. Also, the authors correctly point out that for BCR and TCR reconstruction read length could be reduced. However, the gene expression profile can be significantly impaired by increasing signal to noise ratio, see for example several other analyses performed on bulk RNAseq Conesa, A. et al. A survey of best practices for RNA-seq data analysis. Genome Biol, Changawala et al. The impact of read length on quantification of differentially expressed genes and splice junction detection. Genome Biol 16, as well as Rizzetto et al Scientific Report 2018 on scRNA-seq directly on T cells.

We extended the discussion to point out the disadvantages of short-read sequencing for transcriptomic analysis. We thank the reviewer for their suggestion of references. We note that those papers only evaluate 25bp reads and not 30bp reads, and we were unable to find recent papers which compare 30bp reads to other read lengths.

On another positive note, I am wondering whether this approach could also improve the TCR and BCR detection with the 5' kit from 10X and similar? I have seen several works highlighting that the alignment of the reads from these experiments may be of low quality with standard methods (especially those provided by the companies selling the kits). Perhaps the authors can comment on this as future direction? This tool may be adapted to perform more flexible analyses with a range of technologies?

We thank the reviewer for their suggestion. We implemented an extension of BRAPeS (and TRAPeS) to reconstruct BCR and TCR from 10x data. While our success rates (which are based on reads from the "standard" 10x protocols, and not on VDJ reads generated by the extended T/BCR protocol) are not as high as the 10x VDJ special kit, we do manage to reconstruct hundreds of TCRs and BCRs from 10x data (from the 5' kit and also from 3' data!). We are still working on some problems with this pipeline, but we plan to have a working version of our software for this type of data within a few months.

Reviewer 2:

1.

The authors present the computational tool BRAPeS, an extension of TRAPeS, for reconstruction of B cell receptors (BCRs) from very short reads (25-30 bp). BRAPeS identifies V/J pairs and then reconstructs the CDR3 region in an iterative fashion. The paper demonstrates that BRAPeS is able to reconstruct the CDR3 regions of BCRs from very short reads in a highly sensitive manner, thus opening up the possibility of performing single-cell RNA-sequencing experiments of B-lineage cells using shorter reads than what are currently being used (50-150 bp). Decreased sequencing costs or an increase in number of cells that could be included in a sequencing experiment for the same cost would be very useful for the community.

We thank the reviewer for these remarks.

2.

The data presented support the claim that BRAPeS is more sensitive than existing methods when using very short reads, especially for 25 bp reads and for the heavy chain. Did the authors try to run BASIC and VDJpuzzle using quality-trimmed reads, as they did for BRAPeS, and if so, whether this resulted in any changes in the outcome?

We ran VDJPuzzle and BASIC on the same trimmed reads used by BRAPeS. This resulted in no reconstruction by VDJPuzzle (even for 30bp reads), and a slight decrease in the reconstruction rate for BASIC (33 less productive chains for 30bp data and 8 less productive chains for 25bp data). We have added this note in the materials and methods section, under "Data availability and preprocessing".

The evidence provided by the authors that BCR reconstruction by BRAPeS is accurate is not fully convincing. Firstly, the authors only compared the CDR3 sequences reconstructed by the

various methods, and did not show any data on the accuracy of the remaining parts of the BCR sequences and/or allele assignments.

We thank the reviewer for this very important comment, we have made significant changes to our paper and compared all complementary determining and framework regions, showing the accuracy of BRAPeS across all parts of the BCR transcript (added text to the results section, changed figure 3 and supplementary figure S1, and added supplementary figures S2 and S3)

Secondly, the authors only compared the amino acid sequences of the CDR3s, while this could have easily been performed on the nucleotide level.

We changed our comparison to compare nucleotides instead of amino acids, and the results remained the same.

The accuracy of BCR reconstruction could be better judged by aligning the entire reconstructed BCR nucleotide sequences to the long-read reconstructed BASIC sequences.

We appreciate this comment, we made significant changes the manuscript to also compare the reconstruction accuracy of the CDR1, CDR2 and framework regions. We added more text in the results as well as changed figure 3 and the supplementary figures to show that BRAPeS maintains high accuracy across the different regions of the BCR. We believe that performing a per-region comparison is better than aligning the full reconstructed BCR since it allows a better assessment of the source of a mismatch, in case one exists. In addition, we noticed that BASIC and VDJPuzzle at times report a partial reconstruction (e.g. only a partial FR1 reconstruction, or missing CDR1 sequence) but the correct CDR3, and we think that a partial reconstruction should not be completely discarded as it can still be valuable.

Furthermore, in the Materials and methods section it is stated that «In case of more than one reconstructed CDR3 sequence, if both methods had at least one identical CDR3 sequence it was considered an accurate reconstruction». It would be useful to know how many such cases there were, and whether the identical CDR3 was the most highly ranked by BRAPeS. It should be easy to include these aspects in the comparison of sensitivity and specificity between the methods.

We also compared the accuracy of the methods when taking only the top ranked BCRs of BRAPeS and VDJPuzzle (BASIC resulted in only a single reconstruction per chain). We mention this comparison in the results section and added two supplementary figures. The accuracy remains high, as 97.5% of the CDR3s which are identical between BRAPeS (on both 25bp and 30bp) and BASIC (on long reads) are the top ranked CDR3

While the authors state that BRAPeS corrects for somatic hypermutations in the CDRs, on the GitHub page it is stated that this is currently only true for heavy chain. When will this be implemented for light chains?

We apologize for the delay, the publicly available version of the software now corrects for somatic hypermutations in all chains.

Furthermore, a major drawback of the method is that it does not seem to correct for somatic hypermutations in the framework regions. If the FWR sequences are determined by the assigned V allele, this could be incorrect both due to incorrect assignment of V allele and due to presence of mutations or polymorphisms. Thus, downstream analyses of the reconstructed BCRs may be based on incorrect BCR reconstructions and mutational profiling analyses may be inaccurate. It is also unclear if J segments are corrected for somatic hypermutations, or if they are reconstructed together with the CDR3 after identification of the V/J pair.

We thank the reviewer for pointing out this drawback. We have made changes to BRAPeS to also reconstruct all of the framework regions and changed figure 3 and added text to show that they are accurate reconstructions.

The authors provide no estimations or examples of cost reduction or increase in cell numbers that could be analyzed when reducing the read length from e.g. 50 bp (for which current BCR reconstruction methods work fine) to 25 bp.

As sequencing costs differ between institutions and depend on many factors such as library preparation and/or sequencing machine, we could not find a straightforward answer to this question. The only published paper that compares costs of 25bp vs. longer reads was published in 2015, and shows a difference of ~\$400 in sequencing cost per lane. We also contacted two sequencing facilities - one at UC Berkeley and one at the Chan Zuckerberg biohub. We got various cost estimates, which showed a difference of \$1425-3400 between short (25-37bp) and longer reads (50-75bp). We added a short description of this price range in the discussion, however we believe that a more in depth comparison of the costs between read lengths is beyond the scope of this paper.

Very few scRNA-seq studies currently use 25-30 bp reads, and therefore some evidence on whether shorter reads would influence mapping rate and transcript quantification of the remaining transcriptome need to be provided by the authors.

We added details and references to the discussion. We note that while previous papers have shown increased technical noise in transcriptome analysis for 25bp, we could not find a similar evaluation for 30bp.

3. Lastly, indicate any additional issues you feel should be addressed (text changes, data presentation, statistics etc.).

P=paragraph number in section, L=line

- The authors should provide a small test dataset and expected output files in order to test a working installation of BRAPeS

We have added a small dataset along with expected output files to the GitHub page of BRAPeS.

- The authors should provide information regarding versions of software dependencies that have been tested with BRAPeS

We added details regarding the tested software versions to gitHub

- Abstract:

- o The sentence about measurement of binding specificity needs rewriting since the binding specificity per se is not measured. It should be specified that you mean BCR-mediated specific binding.

We thank the reviewer for the comment, we changed the wording to mention that what we measure is the antigen specificity as determined by the BCR.

- o BRAPeS is publically available at the following link

We thank the reviewer for this (and other) grammar and spelling corrections. We replaced “in” for “at”.

- Introduction:

- o P2 L1: The term IgL is potentially confusing to the reader, as it could be interpreted as the lambda chain

We removed the term IgL from this sentence.

- o P2 L5: "joining segments" should be "joining segment".

We changed the term “joining segments” to “joining segment”.

- o P2 L6: The term "mutations" should be removed as this is covered by insertions and deletions, and may be confused with somatic hypermutations

We removed the term “mutations”

- o P2 L10: Somatic hypermutations also occur in the framework regions, not only CDRs.

Furthermore, class switching does not necessarily occur after activation, so "may be replaced" is a better term.

We changed the sentence to be more accurate as the reviewer suggested.

- o The sentence mentioning 10x genomics looks very messy with a lot of parentheses

We changed the reference style so the sentence will be clearer.

- o Last sentence: Remove "length" from "short length reads"

We removed the word “length”.

- Results:

- o P1: "BRAPeS" is mentioned very often. Please consider to change some of the sentences to passive voice, omitting BRAPeS.

We have changed this paragraph as suggested.

- o P2: Typo: BRAPes

We fixed the typo.

- Materials and methods:

- o The BRAPeS algorithm

- has several steps

We corrected the sentence.

- 3. If the number of bases is the length of the J segment, then won't the iteration in effect start at the 3' end of J and not 5' end? It is unclear to me what is meant by the number of bases in this paragraph.

In this step, BRAPeS align the reads to the V and J segments and allow them to “flank” toward each other, so that we can “fill the gap” between the genomic V and J sequences and reconstruct the CDR3. Since in this step we are only reconstructing the CDR3 region, we only take the bases of each segment that are closest to the start of the CDR3 (i.e. the 3' of the V and 5' of the J). Thus, the number of bases in this paragraph means the number of bases leading up to the edge close to the CDR3. If we take the full J segment you can indeed think of that as starting from the 3' end and we understand how this can be confusing. We have changed the phrasing of this part to better explain our meaning.

- 4.

- Change BCR class to isotype

We changed the term “BCR class” to “BCR isotype”

- Is the CH1 part of C gene used or the entire constant region gene?

The entire constant gene is used, we changed the text to explicitly mention it

- mapped to the

We added “to” to the sentence.

- What about naïve B cells expressing both IgD and IgM?

As part of BRAPeS' output, we provide a file with the read count and TPMs for all isotypes. Thus, a cell which express both IgD and IgM will be easy to detect as it will have high TPM value for both isotypes. We did not previously mention this output file in the manuscript and we thank the reviewer for pointing that out. We added a more detailed description of BRAPeS' output in both the Materials and Method section and in the gitHub page.

- 5. ... optionally extended: Does this mean that the FWRs are optionally reconstructed?

We have extended BRAPeS to always reconstruct the framework regions and changed the text accordingly.

- o Running BRAPeS: Please provide a brief explanation of the parameters that were used.

We provided an explanation of the parameters used. A detailed description of all parameters can be found on the gitHub page.

- o The BRAPeS algorithm

- Figure 1: It is confusing to show alignment to segments in the germline configuration when this is not how the locus looks like in the cells being sequenced. It will be easier to understand if V3 and J6 are not shown, or if it is clearly stated that the segments are in germline configuration.

We have specified both in the figure and the legend that we illustrate the reference genome, and we hope it is clear now

Please change "class determination" to "isotype determination"

We changed the term to “isotype determination”

- Figure 2: Typo, «TCR» should be «BCR»

We fixed the type.

August 11, 2019

RE: Life Science Alliance Manuscript #LSA-2019-00371-TR

Prof Nir Yosef
UC Berkeley

Dear Dr. Yosef,

Thank you for submitting your revised manuscript entitled "Reconstructing B cell receptor sequences from short-read single cell RNA-sequencing with BRAPeS". As you will see, reviewer #2 appreciates the changes introduced in revision and we would thus be happy to publish your paper in Life Science Alliance pending final revisions necessary to meet our formatting guidelines:

- please make sure that the code for the software is available (via github or provided as a zip file)
- we display S figures in-line in the HTML version of the paper -> please upload the supplementary figures as individual files and without legends, please move the legends into the main manuscript file

A. FINAL FILES:

B. MANUSCRIPT ORGANIZATION AND FORMATTING:

Sincerely,

Reviewer #2 (Comments to the Authors (Required)):

The authors have done a great job revising the manuscript and have addressed all my concerns and suggestions with satisfaction.

My main concern was that the authors previously did not demonstrate the accuracy of BCR reconstruction by BRAPeS on the nucleotide level or outside the CDR3 region. The authors have now adequately addressed this by performing a per-region comparison using nucleotide sequences.

Another concern was the lack of documentation regarding the claim that reducing read length would significantly reduce sequencing costs, and whether shorter read length would affect global transcription analyses. The revised manuscript now addresses this concern in a satisfactory manner in the discussion.

August 14, 2019

RE: Life Science Alliance Manuscript #LSA-2019-00371-TRR

Prof Nir Yosef
UC Berkeley

Dear Dr. Yosef,

Thank you for submitting your Methods entitled "Reconstructing B cell receptor sequences from short-read single cell RNA-sequencing with BRAPeS". It is a pleasure to let you know that your manuscript is now accepted for publication in Life Science Alliance. Congratulations on this interesting work.

*****IMPORTANT:** If you will be unreachable at any time, please provide us with the email address of an alternate author. Failure to respond to routine queries may lead to unavoidable delays in publication.*******

DISTRIBUTION OF MATERIALS:

Again, congratulations on a very nice paper. I hope you found the review process to be constructive and are pleased with how the manuscript was handled editorially. We look forward to future exciting submissions from your lab.

Sincerely,

Andrea Leibfried, PhD

Executive Editor
Life Science Alliance
Meyerhofstr. 1
69117 Heidelberg, Germany
t +49 6221 8891 502
e a.leibfried@life-science-alliance.org
www.life-science-alliance.org